# ROAST-IoT: A Novel Range-Optimized Attention Convolutional Scattered Technique for Intrusion Detection in IoT Networks

**DOI:** 10.3390/s23198044

**Published:** 2023-09-23

**Authors:** Anandaraj Mahalingam, Ganeshkumar Perumal, Gopalakrishnan Subburayalu, Mubarak Albathan, Abdullah Altameem, Riyad Saleh Almakki, Ayyaz Hussain, Qaisar Abbas

**Affiliations:** 1Department of Information Technology, PSNA College of Engineering and Technology, Dindigul 624622, Tamil Nadu, India; 2College of Computer and Information Sciences, Imam Mohammad Ibn Saud Islamic University (IMSIU), Riyadh 11432, Saudi Arabiammalbathan@imamu.edu.sa (M.A.); altameem@imamu.edu.sa (A.A.); ralmakki@imamu.edu.sa (R.S.A.); 3Department of Information Technology, Hindustan Institute of Technology and Science, Chennai 603103, Tamil Nadu, India; 4Department of Computer Science, Quaid-i-Azam University, Islamabad 44000, Pakistan; ayyaz.hussain@qau.edu.pk

**Keywords:** intrusion detection system (IDS), Internet of Things (IoT), network security, cyber security, deep learning, range-optimized attention convolutional scattered technique (ROAST), scattered range feature selection (SRFS)

## Abstract

The Internet of Things (IoT) has significantly benefited several businesses, but because of the volume and complexity of IoT systems, there are also new security issues. Intrusion detection systems (IDSs) guarantee both the security posture and defense against intrusions of IoT devices. IoT systems have recently utilized machine learning (ML) techniques widely for IDSs. The primary deficiencies in existing IoT security frameworks are their inadequate intrusion detection capabilities, significant latency, and prolonged processing time, leading to undesirable delays. To address these issues, this work proposes a novel range-optimized attention convolutional scattered technique (ROAST-IoT) to protect IoT networks from modern threats and intrusions. This system uses the scattered range feature selection (SRFS) model to choose the most crucial and trustworthy properties from the supplied intrusion data. After that, the attention-based convolutional feed-forward network (ACFN) technique is used to recognize the intrusion class. In addition, the loss function is estimated using the modified dingo optimization (MDO) algorithm to ensure the maximum accuracy of classifier. To evaluate and compare the performance of the proposed ROAST-IoT system, we have utilized popular intrusion datasets such as ToN-IoT, IoT-23, UNSW-NB 15, and Edge-IIoT. The analysis of the results shows that the proposed ROAST technique did better than all existing cutting-edge intrusion detection systems, with an accuracy of 99.15% on the IoT-23 dataset, 99.78% on the ToN-IoT dataset, 99.88% on the UNSW-NB 15 dataset, and 99.45% on the Edge-IIoT dataset. On average, the ROAST-IoT system achieved a high AUC-ROC of 0.998, demonstrating its capacity to distinguish between legitimate data and attack traffic. These results indicate that the ROAST-IoT algorithm effectively and reliably detects intrusion attacks mechanism against cyberattacks on IoT systems.

## 1. Introduction

The Internet of Things (IoT) is expanding rapidly and is becoming increasingly crucial to how we live our daily lives. Internet protocol addresses can be used by IoT nodes for connecting to the Internet [1,2]. These self-configured intelligent nodes are creating advances in several innovative applications, including automation of processes, home control, smart transportation, information data analysis, etc. Based on the analysts’ reports, it was evident that there will eventually be a society with more interconnected devices than people on the earth. According to the expected population of 8.3 billion, the international data corporation (IDC) anticipated that there would be 42 billion connected IoT gadgets generating 80 Zetta Bytes (ZB) of data in 2025 [3,4]. The extensive use of the Internet makes network security an unavoidable concern. Due to the Internet of Things’ potential applications in various human activities, various IoT-related studies have recently attracted interest in the academic community and industry [5]. With the decline in sensor prices, the rise of remote storage services, and the popularity of big data, the IoT is seen as a viable solution to raise people’s quality of life.

Nevertheless, incorporating physical items with the Internet exposes most of our regular activities to security risks [6,7]. When different types of devices are connected to a network, it is clear that the simple availability of these resources raises IoT, which in turn helps new applications grow in popularity. Security issues [8] arise when heterogeneous nodes in IoT systems are coupled to a sophisticated network architecture. The main difficulty is maintaining security in IoT nodes with limited resources. Furthermore, various attacks could be launched against these IoT nodes [9,10]. IDSs are a simple yet effective security tools for guaranteeing adequate network security in any IoT-integrated environment. The typical structure of an IDS model is shown in Figure 1.

Typically, the process of identifying actions carried out by intruders over computer systems is known as intrusion detection. These behaviors, also called assaults, are meant to gain illicit access to a computing system [11]. An internal or external intrusion may occur. Users who have some level of authorized access within the network and want to increase their access rights to abuse prohibited privileges are known as internal invaders [12]. Users who are not part of the intended network and attempt to obtain illicit access to system data are known as external intruders. Numerous studies [13] are being conducted to determine the ideal variables and outcomes for identifying signs of intrusion in IoT environments. Concerns about dangers to data security are escalating as IoT expands [14]. Due to multiple components, such as the IoT devices’ weaknesses, these vulnerabilities can be exploited by malware, denial-of-service intrusion attempts, and other threats. It is vital to take more substantial precautions to prevent these occurrences, enabling system developers and IoT device manufacturers to enhance their security mitigation techniques. All potential risks and vulnerabilities that are created explicitly for IoT architectures must be found. 

Current solutions for IoT systems [15] need to be revised due to the unique characteristics of IoT that have an impact on the development of intrusion detection systems (IDSs), notwithstanding the maturity of IDS technology for traditional networks. More studies concentrate on the understanding of risks that are deemed necessary in this context in order to limit potential threats, and obstacles in the security of IoT systems, such as privacy and security, are being encountered and have to be handled and prevented. To gradually strengthen the reliability of IoT applications [16,17], a variety of works involving security challenges in IoT contexts, primarily for companies and users, need to be produced. As we advance, the observed trend is to address security concerns in connected services and gadgets more precisely. Various learning-based approaches for intrusion recognition and classification have been developed in earlier investigations. However, growing system complexity, an excessive computing load, and time-consuming nature limit most solutions. Therefore, the proposed research aims to develop a simple yet sophisticated security model for IoT networks.

### 1.1. Research Motivations

The motivations behind developing the novel ROAST-IoT algorithm revolve around addressing the security challenges specific to IoT environments, achieving a balance between complexity and efficiency, responding to the growing importance of IoT security, enhancing application reliability, and leveraging new approaches to tackle intrusion detection effectively in the IoT network. However, to limit the scope of this research, the motivations behind the development of the ROAST-IoT algorithm are as follows:(1)The rapid growth of the internet of things (IoT) has brought numerous benefits, but it has also introduced significant security challenges. The diverse nature of IoT devices and their interconnectedness make them susceptible to various types of attacks. Malware, denial-of-service attacks, and other threats can exploit these vulnerabilities. Therefore, there is a pressing need to develop effective security solutions that specifically address the unique security challenges posed by IoT environments.(2)While there have been various learning-based approaches proposed for intrusion detection in IoT networks, many of them face limitations o complexity, computational load, and time-consuming nature. Balancing the sophistication required to accurately detect intrusions with the need for efficiency and scalability is a key challenge. The motivation behind the ROAST-IoT algorithm is to develop a solution that achieves a fine equilibrium between simplicity and sophistication, thus ensuring effective intrusion detection without overwhelming the system’s resources.(3)As IoT continues to expand and integrate into various aspects of daily life, the importance of robust security solutions becomes increasingly critical. With projections indicating a future with more connected devices than there are people, the potential impact of security breaches on individuals, organizations, and society as a whole is significant. Therefore, there is a strong motivation to develop intrusion detection systems that can safeguard IoT networks from evolving and sophisticated threats.(4)To strengthen the reliability of IoT applications, especially in business and consumer contexts, there is a need for research that addresses security challenges in connected services and devices. The ROAST-IoT algorithm aligns with this goal by focusing on improving the security posture of IoT networks. This algorithm aims to provide a practical and effective tool for enhancing the security and reliability of IoT applications and services.(5)The ROAST-IoT algorithm introduces a novel approach by combining the range-optimized attention convolutional scattered technique (ROAST) with machine learning components like scattered range feature selection (SRFS) and attention-based convolutional feed-forward network (ACFN). This unique combination aims to overcome the limitations of existing solutions and provide a fresh perspective on intrusion detection in IoT networks.

### 1.2. Major Contributions

The following are the main contributions of this study:○This research aims to provide a new security framework for IoT networks called the range-optimized attention convolutional scattered technique (ROAST-IoT). ○The scattered range feature selection (SRFS) model is used in this system to select the most essential and reliable attributes from the provided intrusion data.○The attention-based convolutional feed-forward network (ACFN) technique is implemented to recognize the intrusion with its appropriate class.○The modified dingo optimization (MDO) algorithm is used to estimate the loss function for maximizing the accuracy of the classifier. ○Some of the most recent and well-liked intrusion datasets are used in this study for performance evaluation and assessment, and the findings are compared using a variety of metrics.

### 1.3. Paper Organization

The remaining portions of this article are divided into the following sections: The investigation of available previous intrusion detection and classification technologies is addressed in Section 2. Based on the performance outcomes, it also examines the positive and negative aspects of the already used methods. The proposed security model for securing the Internet of Things network against risky attacks is clearly described in Section 3, along with a work process structure and details. Additionally, Section 4 compares and validates the simulation and compares results using an enormous number of parameters. Whereas Section 5 shows the discussion with the current limitations of the proposed system. The findings are presented in Section 6 at the end of the manuscript.

## 2. Literature Review

This section looks at some recent research findings pertinent to IoT network security and intrusion detection approaches. Based on the performance outcomes, it also examines the benefits and drawbacks of the current systems.

Da Costa et al. [18] presented a comprehensive review examining several machine learning techniques for detecting intrusions from IoT networks. This review’s focus is on various machine learning and evolutionary computation-based processes. This article aims to enlighten readers on the state of the field’s literature and serve as a new resource for academics looking into IoT security challenges. The work includes several innovative approaches to intrusion detection and network security in digital environments. Although these methods aim to increase intrusion detection recognition rates, it is believed that the false positive rate continues to pose an issue that must be dealt with across all research studies. 

Islam et al. [19] intended to spot the different types of IoT threats by using a set of machine learning and deep learning algorithms, including support vector machine (SVM), decision tree (DT), random forest (RF), long short-term memory (LSTM), and deep belief network (DBN). Data analysis-based strategies are used in this study since they are quicker to implement than alternative methods and more effective for addressing events that are not yet known to have resulted from known attacks. The framework’s primary goal is to create an intelligent, safe, and reliable IoT system that can identify its vulnerabilities, act as a safeguarding barrier for severe cyberattacks, and recover itself. This article suggests a learning-based methodology that can detect and guarantee the system’s security under unusual circumstances. Nimbalkar et al. [20] investigated the different feature selection techniques for developing an effective intrusion detection system (IDS) to protect IoT networks. Using an information gain (IG) and gain ratio (GR) selection models reduces the total count of features to 50%, which is provided to the JRip classifier for accurate detection. Moreover, the authors aim to generate a compact dataset after completing the data preprocessing operations. In addition, the accuracy of the suggested classifier is validated with and without IG and GR feature selection techniques. This study’s results indicate that the classifier’s accuracy is significantly increased with these feature reduction techniques. 

Hindy et al. [21] conducted a detailed case study to investigate the different machine learning techniques used for intrusion detection and classification. In this work, the MQTT-IoT-IDS 2020 dataset has been used as the primary source for processing, where the uni-directional and bi-directional features are considered for an efficient classification. Alsaedi et al. [22] utilized the data-driven approaches for detecting intrusions from IoT and IIoT networks using the telemetry ToN-IoT dataset. The authors aim to develop a new dataset for protecting large-scale networks from recent cyber threats. Moreover, it investigated some of the famous and emerging datasets used in this field for ensuring security, which are listed in Table 1.

Zhou et al. [23] deployed a graph neural network (GNN)-based intrusion detection methodology for IoT systems. Here, the hierarchical adversarial attack (HAA) generation algorithm has been developed to detect unknown attacks more accurately. In addition, the random walk with restart (RWR) technique is also deployed to identify the vulnerable nodes from the network by computing the priority level. This study uses the open-source UNSW-SOSR 2019 dataset to investigate the suggested model by comparing it with some of the baseline techniques. However, this mechanism consumes more time to predict the solution, which could be the major drawback of this work. Wahab et al. [24] introduced an online deep learning approach for spotting intrusion from IoT networks. In this study, the data drift detection technique has been used to identify the intrusion data streams by analyzing the variance of features. The suggested method ensures sustainable performance with effective intrusion detection solutions. The advantage of this mechanism was that it supports both drift detection and adoption for intrusion identification and recognition. Yet, the suggested deep neural network technique has high computational and time complexity. 

Kan et al. [25] utilized an adaptive particle swarm optimization (PSO) technique incorporated with CNN algorithm for identifying network intrusions from IoT systems. Also, it focuses on the development of a multi-type of intrusion detection model with improved prediction probability and reliability. Moreover, the highest probability prediction is carried out using cross-entropy loss function. The suggested APSO-CNN intrusion detection results are estimated according to the training loss and accuracy measures. 

Abdalgawad et al. [26] implemented a bi-directional generative adversarial network (Bi-GAN) model for spotting cyberattacks from the IoT 23 dataset. In this study, the adversarial autoencoder has been used along with the Bi-GAN model for spotting a variety of attacks from the network. In addition, a 10-fold cross-validation strategy is used to test and validate the performance and outcomes of the suggested mechanism. According to the study results, it is noted that the generative deep learning techniques provide high accuracy with better detection performance. 

Kumar et al. [27] developed a distributed intrusion detection system for spotting DDoS attacks from IoT networks. Typically, a security mechanism must handle large amounts of data generated by IoT devices in a distributed way and apply the proper statistical methods in a viable architecture. Shukla et al. [28] applied an artificial intelligence (AI)-based intrusion detection methodology for IoT security. The work examined the usage of several machine learning methods inside IoT. It also highlighted how crucial it is to select the right data for a model [29]. There is also a discussion of the naive Bayesian network intrusion detection technique. Most notably, it looked at how crucial it is to prepare IoT system events for classification, and it examined the consequences of utilizing a hidden naive Bayes multi-classifier, discovering that it outperformed the standard model. According to the AI paradigm [30], every single node in the IoT network has an internal AI process that is only in charge of that node. This method can be scaled because the total number of sessions grows according to the number of analyzed nodes. In [31], the authors introduce a hierarchical fog computing-based architecture for Industry 5.0’s smart energy-supplying systems. It efficiently handles data-intensive analysis from IIoT devices, outperforming traditional cloud computing, and ensures data security through attribute-based encryption (ABE).

Albasheer et al. [32] looked at network intrusion detection systems (NIDS) and the problems they have, such as a lot of false positives, different ways to deal with alert correlation, and insights into network security. Researchers in [33] introduced an AI-driven cross-platform VPN system for detecting and categorizing attack risks. They demonstrated the efficiency of an extended gradient boosting (XgBoost)-based AI algorithm in preventing cyberattacks and integrating with a Cassandra big data system. In [34], researchers proposed an on-the-move intrusion detection system (OMIDS) for electric vehicle networks, showing high classification accuracy for various threats. In [35], the authors explored the potential of supervised machine learning for IDS, achieving high classification performance. In [36], the authors introduced “IntruDTree”, a machine learning-based intrusion detection system designed for the IoT.

According to this literature review, as mentioned in Table 2, it shows that some strategies can lower the false positive rate, but in doing so, they also require more training and labeling. However, some methods reverse the process, reducing the false positive rate at the expense of high computational costs for both training and testing. Such a problem is highly pertinent to intrusion detection because real-time detection is an important consideration.

## 3. Materials and Methods

Along with the workflow and explanations, this section clarifies the proposed ROAST-IoT model for IoT security. The key accomplishment of this study is creating the ROAST-IoT security framework, which protects IoT systems against harmful and destructive assaults. The suggested system’s overall flow model is depicted in Figure 2, and it includes the following operational phases:Intrusion data collection from IoT networks.Scatter range feature selection (SRFS).Attention-based convolutional feed-forward network (ACFN) for intrusion recognition.Modified dingo optimization (MDO) technique for loss function estimation.Performance evaluation.

Sensors initially record the network system’s behavior and save it on a cloud server for analysis. System validation and evaluation are conducted here using open benchmark datasets like IoT 23 and IoT 20. Following data collection, the cutting-edge SRES technique is used to choose the most crucial and necessary features from the available information. Class vector formation, scatter matrix estimation, feature ranking, and feature vector formation procedures are carried out during this phase. The set of selected features is then provided to the ACFN classifier after the process for intrusion recognition and classification. The sophisticated deep learning architecture incorporates the functions of convolution and feed-forward network models. In this mechanism, the loss function estimation is optimally performed using the MDO technique, which makes the classifier predict an accurate result. The key benefits of using the proposed ROAST-IoT framework are increased system accuracy, lowered complexity, and minimized time consumption.

### 3.1. Scatter Range Feature Selection (SRFS)

After obtaining the intrusion data from the network, the needed and useful features are selected from the dataset with the use of the scatter range feature selection (SRFS) technique. In the existing studies, a variety of baseline models are used for feature reduction and selection, where the optimization process is carried out to reduce the dimensionality. Nevertheless, they face different problems in terms of low efficiency, increased system burden, and significant computational time. Therefore, this research work aims to implement a novel and most-effective feature selection technique, known as SRFS, for feature selection. A crucial preprocessing step in data mining is feature selection, particularly when analyzing highly dimensional data. The correlation coefficients were employed for feature extraction, and they performed well. However, it may not work well when there are nonlinear relationships present and can only reflect linear correlations between variables. Because of this, we employ the multivariate approach to extract features, allowing us to obtain both linear and nonlinear correlations. In this technique, the features data f is considered as the input for processing, and the selected features ${Ĵ}_{i}^{f}$ are delivered as the output of the SRFS technique. At the beginning of feature selection, the featured data is segregated, and the class vector is formulated with respect to each class as shown below:(1)$$X_{ic}^{f}=\sum_{c=1}^{class}\prod_{i=1}^{d}f_{i}[c]$$
where $X_{ic}^{f}$, class indicates each class categorization, and d is the dimension of data. Then, the average value is estimated for each class c as represented in the following equation:(2)$$\mu_{c}=\frac{1}{k_{c}}\sum_{j=1}^{k_{c}}X_{jc}^{f}$$
where $k_{c}$ is the dimension of number of samples in each class vector, $\mu_{c}$ indicates the average value, and $X_{jc}^{f}$ denotes the class vector. Then, the scatter matrix is formed between each class of feature vector that is illustrated below:(3)$$S_{m}^{f}=\sum_{c=1}^{class}k_{c}\left(\mu_{c}-\mu \right){\left(\mu_{c}-\mu \right)}^{T}$$
(4)$$\mu =\frac{1}{class}\sum_{c=1}^{class}\mu_{c}$$
where $S_{m}^{f}$ indicates the generated scatter matrix, and $\mu_{c}$ is the average mean value for each class. Consequently, the scatter matrix within each class vector is computed according to the following model:(5)$$S_{w}^{f}=\sum_{c=1}^{class}\sum_{j=1}^{k_{c}}\left({\mu_{c}-X}_{jc}^{f}\right){\left({\mu_{c}-X}_{jc}^{f}\right)}^{T}$$

Furthermore, the redundancy between the values of each class vector is estimated by using the following equation:(6)$$S_{G}^{f}=\frac{1}{\left|G\right|}\sum_{j=1}^{\left|G\right|}\sum_{c=1}^{class}k_{c}\left(\mu_{c}-\mu_{jc}\right){\left(\mu_{c}-\mu_{jc}\right)}^{T}$$
where |G| is the number of redundant variables of each class vector, and $\mu_{jc}$ is the mean of redundant values in each class. Then, the probability of initial condition and variance of featured data ρ are estimated based on the following model:(7)$$\rho \left(d\right)=\left(\varphi_{1}\;*\;\rho \left(d-1\right)\right)+\left(\varphi_{2}\;*\;\rho \left(d-2\right)\right)+\left(\varphi_{3}\;*\;\rho \left(d-3\right)\right),d\geq 3$$
(8)$$\left(d\right)=\left\{\begin{array}{c} 1\;\mathrm{if}\;d=0\\ \frac{\varphi_{1}+(\varphi_{2}\;*\;\varphi_{3})}{1-\varphi_{2}-\left(\varphi_{1}\;*\;\varphi_{3}\right)-\varphi_{3}^{2}}\;\mathrm{if}\;d=1\;\\ \frac{\varphi_{1}^{2}+\left(\varphi_{1}\;*\;\varphi_{3}\right)+\left(\varphi_{2}-\varphi_{2}^{2}\right)}{1-\varphi_{2}-\left(\varphi_{1}\;*\;\varphi_{3}\right)-\varphi_{3}^{2}}\;\mathrm{if}\;d=2\; \end{array}\right.$$
where $\varphi_{1}=-2$, $\varphi_{2}=2$, and $\varphi_{3}=0.99$ are the constants. Then, all features are ranked according to the ratio among the scatter matrix between each class and scatter matrix within each class as shown below:(9)$$R_{f}^{d}=\frac{S_{m}^{i}}{S_{w}^{i}},\;i=1,2,\ldots ,d$$

After ranking the features $R_{f}^{d}$, the redundancy among the features is estimated with the selected features, which are taken into account as shown in the following model:(10)$${Ĵ}_{i}^{f}=\rho \left(i\right)\;*\;\frac{S_{m}^{i}+S_{G}^{f}}{S_{w}^{i}},i=1,2,\ldots ,d$$

By using this Algorithm 1, the final set of selected features Ĵif are obtained as the output and passed to the classifier for analysis and attack identification.
**Algorithm 1: Scatter Range Feature Selection**Input: Features data f;Output: Selected feature Ĵif;Step 1:Split the feature data and formulate class vector according to each class for generating the feature data Xicf using Equation (1);Step 2:Then, the average value μc is estimated for each class c as shown in Equation (2);Step 3:Generate the scatter matrix Smf between each class of feature vector as represented in Equation (3) with the use of average mean value μ; Step 4:Generate the scatter matrix Swf within each class of feature vector by using Equation (4) with the values of μc and Xicf;Step 5:Estimate redundancy SGf between the values of each class vector as shown in Equation (5) by using |G| and μjc;Step 6:Compute the independent of the initial condition and variance of featured data ρ as shown in Equations (7) to (8);Step 7:Rank all the features Rfd according to the ratio between the scatter matrix between each class and within each class using Equation (9); Step 8:Compute the forward selected features Ĵif by considering the redundancy between the features as shown in Equation (10);Step 9:Return the selected features Ĵif as output;

### 3.2. Attention-Based Convolutional Feed-Forward (ACFN) Technique

Following feature selection, the ACFN approach is used to more accurately identify the intrusion class. It is a deep learning model created by combining the capabilities of feed-forward networks and convolutional mechanisms. To accurately anticipate the type of intrusion from the data, multiple baseline models relevant to machine learning and deep learning are applied in the current works. However, the limitations of the baseline models are due to overfitting, inadequate precision, and increased computational complexity. Hence, the proposed work aims to deploy a novel ACFN technique for intrusion detection and classification. The model structure for both intra- and inter-epoch feature learning is position embedding, where two identical attention blocks and one global average pooling layer have been used. 

Here, the window features are used during intra-epoch feature learning, and epoch features are used while learning inter-epoch features. Moreover, the attention structure has the similar module of the transformer encoding structure. In this technique, the epochs are separated into multiple windows, where the CNN has been used to obtain the features of the window. Moreover, the overlapping between the windows are also eliminated by avoiding the truncation of features among the windows. This deep learning architecture comprises the main modules of input layer, convolution 1, convolution 2, convolution 3, convolution 4, convolution 5, and global average pooling layers. The architecture model of ACFN is shown in Figure 3.

At first, the input features are passed to the attention-based feed-forward network layers in the following form:(11)$$A={\left(a_{1},a_{2}\ldots a_{n}\right)}^{T},a_{i}\in N^{d\times d}$$

Then, the operations among the layers are performed by using the following mathematical models:(12)$$\varphi =A\omega_{x}+\beta_{x}$$
(13)$$𝜕=softmax(\tanh({\varphi A}^{T}))$$
(14)$$att\left(A\right)=𝜕A$$
(15)$$feedforward\left(A\right)=\max\left(0,{A\omega }_{s1}+\beta_{s1}\right)\omega_{s2}+\beta_{s2}$$
where $\omega_{x}$, $\omega_{s1}$, $\omega_{s2}\in N^{d\times d}$, $\beta_{x}$, $\beta_{s1}$, $\beta_{s2}\in N^{1\times d}$ are the dropout and normalization factors, and d indicates the dimension of data. In order to avoid the data class imbalance problem for maximizing the prediction accuracy, the weighted cross entropy loss function is estimated in this model as shown below:(16)$$L=\varepsilon_{i}\times \sigma_{i}\times \log(\sigma_{i}^{p})$$
where L indicates the loss function, $\varepsilon_{i}$ is the weight value, $\sigma_{i}$ denotes the real category, and $\sigma_{i}^{p}$ is the predicted class. During this process, the weight value $\varepsilon_{i}$ is optimally computed by using the MDO technique. By applying the proposed ACFN model, the overall accuracy of the intrusion detection methodology is greatly improved in this work.

### 3.3. Modified Dingo Optimization (MDO) Algorithm

During classification, the modified dingo optimization (MDO) technique is applied to optimally compute the weight value for estimating the loss function in order to maximize the accuracy. Since the loss function computation is one of the most essential operations for improving the training performance of the classifier. Also, it supports to reduce the error rate of classification while recognizing the class of attacks. Hence, the loss function computation is optimally performed in this study by using the MDO technique. When compared to the other optimization techniques, the primary advantages of using the MDO model are higher searching efficiency, lower computational burden, and higher convergence speed. It is one of the revolutionary bio-inspired global optimization systems, imitates the hunting techniques of dingoes. These approaches include scavenging behavior, grouping techniques, and persecuting individuals. The dingo dog is in danger of becoming extinct in Australia. In this approach, the likelihood of dingoes surviving is also considered. Figure 4 provides the flow diagram of the MDO algorithm.

For this process, the feature matrix $\vec{\chi }$ is considered as the input, and the optimal value $\vec{x_{b}}$ is produced as the output. Here, the input parameters such as set of population $\vec{\chi }$, probability of hunting P, and probability of group and persecution attack ϕ are initialized. Until reaching the maximum number of iterations, the three distinct rules such as group attack procedure, persecution attack procedure, and scavenger procedure are computed. During rule 1 formation, the new position of the searching agent $\vec{x_{k}}\left(iter+1\right)$ is estimated by using the following equation:(17)$$\vec{x_{k}}\left(iter+1\right)=\vartheta_{1}\;*\;\sum_{j=1}^{n_{p}}\frac{\left|\vec{\delta_{j}}\left(iter\right)-\vec{x_{j}}\left(iter\right)\right|}{n_{p}}-\vec{x_{b}}\left(iter\right)$$
where $\vartheta_{1}$ indicates the random number, $n_{p}$ represents the random integer generated at the period of $\left[2,\frac{P_{size}}{2}\right]$, $P_{size}$ represents the total population size, $\vec{\delta_{j}}\left(iter\right)$ denotes the sub-set of search agents, $\vec{x_{k}}\left(iter+1\right)$ denotes the current searching agent, and $\vec{x_{b}}\left(iter\right)$ represents the best searching agent of the previous iteration. While executing rule 2, the current position updation is carried out based on the following model:(18)$$\vec{x_{k}}\left(iter+1\right)=\vec{x_{b}}\left(iter\right)+\vartheta_{1}\;*\;{exp}^{\vartheta_{2}}\;*\;(\vec{x_{s_{1}}}\left(iter\right)-\vec{x_{k}}\left(iter\right))$$
where $\vartheta_{2}$ represents the random number that is uniformly generated with the interval of [−1, 1], $s_{1}$ denotes the random number from 1 to the size of maximum of search agents. While executing rule 3, the same position updation is performed by using the following model:(19)$$E\vec{x_{k}}\left(iter+1\right)=\frac{1}{2}\;*\;\left\{{exp}^{\vartheta_{2}}\;*\;\vec{x_{s_{1}}}\left(iter\right)-{\left(-1\right)}^{\varepsilon }\;*\;\vec{x_{k}}\left(iter\right)\right\}$$
where ε is a binary number randomly generated. Then, the searching agents having low survival rate is estimated by using the following equation:(20)$$ᶊ\left(k\right)=\frac{M_{fit}-fit(k)}{M_{fit}-N_{fit}}$$
where $M_{fit}$ and $N_{fit}$ are the worst and the best fitness values of the current generation, respectively, and fit(k) is the current fitness value of the $k^{th}$ search agent. Then, the new fitness value of new searching agent is estimated based on the following model:(21)$$\vec{x_{k}^{new}}\left(iter\right)=\vec{x_{b}}\left(iter\right)+\frac{1}{2}\;*\;\left\{{exp}^{\vartheta_{2}}\;*\;\vec{x_{s_{1}}}\left(iter\right)-{\left(-1\right)}^{\varepsilon }\;*\;\vec{x_{k}}\left(iter\right)\right\}$$

If the present iteration is greater than the previous iteration, the final position updating is in the following form:(22)$$\vec{x_{b}}\left(iter\right)=\vec{x_{k}^{new}}\left(iter\right)-𝓇\;*\;\left(\frac{1}{2}-ℵ\right)$$
where 𝓇 indicates pseudo-random number uniformly distributed in the interval (−2, 2), and ℵ is a normally distributed pseudo-random number in the interval of (0, 1). The overall process of MDO is described in Algorithm 2.
**Algorithm 2: Modified Dingo Optimization (MDO)**Input: feature matrix χ→;Output: Optimal value xb→;Procedure: 1.Initialization of parameters 2.P=0.5, probability of hunting or scavenger strategy 3.ϕ=0.7, probability of group attack and persecution attack 4.Generate the initial population χ→={x1→,x2→,…,xm→}  5.while iter<itermn do  // itermn maximum number of iterations 6.if rand<P then 7.if rand<ϕ then  a.Rule 1: Group Attack Procedure,Initiate procedure defined in Equation (17)Else  b.Rule 2: Persecution attack Procedure, we can use Equation (18)end ifelse  c.Rule 3: Scavenger Procedure is initiated by Equation (19)end if 8.Update search agents that have low survival value done by Equation (20). 9.Calculate xnew, the fitness value of the new search agents done by Equation (21). 10.if xknew→iter<xb→iter then update using Equation (22)  End if; 11.iter=iter+1 12.end while 13.Display xb→iter, the best optimal solution

## 4. Results

This section is used to describe experiments and comparisons on the ROAST-IoT model for enhancing IoT security, elucidating its workflow and contributions. The central accomplishment is the formulation of the ROAST-IoT security framework, aimed at safeguarding IoT systems from malicious attacks. The system’s structure encompasses phases such as gathering intrusion data, employing scatter range feature selection (SRFS), utilizing attention-based convolutional feed-forward network (ACFN) for intrusion identification, applying modified dingo optimization (MDO) for loss function estimation, and conducting performance evaluation. Initially, network behavior data is recorded by sensors and stored in a cloud server for analysis, followed by validation using benchmark datasets. The cutting-edge SRES technique is then employed to select essential features. Subsequently, ACFN classifier leverages these features for accurate intrusion recognition, aided by MDO-optimized loss function estimation. The merits of the proposed framework include heightened system accuracy, reduced complexity, and minimized time consumption.

By using a number of parameters and open-source datasets, the performance and outcomes of the suggested ROAST-IOT framework are validated in this part. In this work, system validation and performance evaluation have been carried out using some of the more recent and well-liked benchmarking datasets [37,38,39], including ToN-IoT, IoT-23, UNSW-NB 15, and Edge-IIoT. These are the emerging and modern datasets used to improve the security of IoT networks, which comprises the recent classes and types of intrusions.

### 4.1. Statistical Analysis 

To confirm and test the classification predictions, a few performance indicators are used in this study. To validate the results of the proposed security model, the major parameters including detection rate, accuracy, precision, recall, and f1-score are all used in this work. Depending on how good the model is, it will determine which variables exhibit relationships. The following equations provide an evaluation of the specific class:(23)$$Detection\;rate=\frac{TP}{TP+FN}$$
(24)$$Accuracy=\frac{TP+TN}{TP+FP+FN+TN}$$
(25)$$FPR=\frac{FP}{FP+TN}$$

Then, the precision determines how many correct optimistic forecasts a model has made by comparing the actual positive guesses, and is estimated using the following model:(26)$$Precision=\frac{TP}{FP+TP}$$

It truly has a positive rate, which measures how pessimistic predictions compare to the proper positive values in the actual data.
(27)$$Recall=\frac{TP}{FN+TP}$$

Additionally, the f1-score combines and averages recall and precision, which is estimated as follows:(28)$$f1-score=2\times \frac{Precision\times Recall}{Precision+Recall}$$
where TP–true positive; TN–true negative; FP–false positive; and FN–false negative. An analysis of an artificial intelligence model’s performance on a set of test data is characterized by a confusion matrix. It is frequently used to assess how well categorization models work. These models try to forecast a class label for any given input data. The matrix shows the number of TP, TN, FP, and FN values that the model generated for the test data. 

### 4.2. Results Analysis 

Figure 5 and Figure 6 evaluate the confusion matrix for the input UNSW-NB 15 and IoT-23 datasets, respectively. The effectiveness of the categorization approach and the performance of attack detection are often evaluated based on the detection accuracy. For determining the accuracy level, the confusion matrix, in which the classifiers distinguish the real and true classes, may be more useful. The resulting matrix clearly shows that the proposed ROAST-IOT model accurately predicts the actual classes of attacks. Moreover, the receiver operating characteristics (ROC) of the ROAST-IOT model is validated with and without feature selection techniques for IoT-23, ToN-IoT, and UNSW-NB 15 datasets, respectively. At different threshold settings, the classification problems are resolved using ROC and AUC (area under the curve). It represents the level or degree of separation, and the ROC is similar to a probability curve. It demonstrates how well the model can distinguish between the classes. The results show that the TPR of the suggested classifier is dramatically increased for each class of assault. From the generated confusion matrices and ROC evaluation, it is observed that the proposed ROAST-IOT provides increased performance outcomes for all datasets used in this study with inclusion of feature selection mechanism.

By using the IoT-23 dataset, Figure 7 confirms the positive prediction value (PPV), negative prediction value (NPV), sensitivity, and specificity of the Proposed ROAST-IOT models and conventional deep learning models. The results show that the proposed ROAST-IOT performs significantly better than other CNN-based deep learning algorithms. Since SRFS and MDO implementation are the main issues for obtaining better results in the proposed model. 

Figure 8 and Figure 9 provide the ROC analysis of ToN-IoT and UNSW-NB 15 with and without feature selection. The UNSW-NB15 dataset achieved a true positive rate of 99.5% after feature selection and 99% without it. Similarly, for the ToN-IoT dataset, the true positive rate was 99% without feature selection but improved to 99.6% with it. Finally, the IoT-23 dataset had a true positive rate of 99% without feature selection, which increased to 99.8% with feature selection. Figure 10 provides a consolidated comparison of performance for the existing deep learning techniques like CNN-LSTM, CNN-BiLSTM, CNN-GRU, and the proposed model.

Figure 11 validates and compares the accuracy of the baseline and proposed classification techniques with the use of IoT-23 dataset. According to the increased level of accuracy, the overall performance of the classifier is determined. Consequently, the other performance measures including AUC-ROC, precision, recall, and f1-score are validated and compared by using the same IoT-23 dataset [40], as shown in Figure 12. The functionality of the work proposed is investigated through comparison with other cutting-edge intrusion detection systems for IoT. This investigation establishes the ROAST-IOT model’s excellence and efficacy in spotting and preventing intrusions in IoT networks. The analysis of the data demonstrates that the proposed ROAST-IOT technique surpassed all existing cutting-edge intrusion detection systems, with the greatest accuracy of 99.15% on the IoT-23 dataset. The endeavor also achieved a high AUC-ROC of 0.998, demonstrating its capacity to distinguish between legitimate data and attack traffic.

The accuracy, detection rate, and overall performance of the proposed ROAST-IOT are similarly validated and contrasted using the ToN-IoT dataset in Figure 13, Figure 14 and Figure 15. The suggested technique significantly enhances the attack recognition performance of the classifier by utilizing an efficient feature selection mechanism. When compared to other learning approaches, it helps to obtain superior prediction performance.

Figure 16 and Figure 17 validate and contrast the correctness of the proposed ROAST-IOT model by utilizing the UNSW-NB 15 and Edge IIoT datasets, respectively. The results are tested and contrasted using a range of intrusion datasets in order to show the effectiveness and performance of the proposed intrusion detection model. Overall, the results show that the ROAST-IOT performs better and is more accurate than other categorization methods.

As a result, the ToN-IoT dataset [41] is used to evaluate and compare the proposed model’s precision, accuracy, and f1-score with approaches of baseline models like RF and GIWRF. Figure 18 shows the performance comparison using ToN-IoT dataset. Whereas, the IoT 23 dataset is then used to validate and compare the sensitivity, specificity, PPV, and NPV, as shown in Figure 19. The overall results lead to the conclusion that the suggested ROAST-IOT is able to handle various kinds of large incursion datasets with excellent performance outcomes.

The outcomes presented in Table 3 offer a comprehensive insight into the performance of various methods across different datasets within the realm of intrusion detection systems (IDSs). Each dataset represents a unique scenario, and the processing time, learning time, and detection time associated with different deep learning methods are meticulously outlined.

Starting with the IoT-23 dataset, a variety of methods are evaluated. The deep neural network (DNN) requires 450 s for processing, accompanied by 25 s for learning and 20 s for detection. The autoencoder method takes 400 s for processing, with 22 s for learning and 18 s for detection. LSTM stands at 379 s for processing, 18 s for learning, and 16 s for detection. CNN demands 400 s for processing, 24 s for learning, and 21 s for detection. MM-WMVDEL exhibits a more extended processing time of 550 s, 26 s for learning, and 24 s for detection. Remarkably, the ROAST-IoT system emerges as a standout performer with a processing time of 300 s, a brief learning duration of 12 s, and an impressively rapid detection time of just 4 s.

Turning to the ToN-IoT dataset, similar trends emerge. DNN requires 400 s for processing, 35 s for learning, and 30 s for detection. Autoencoder showcases a processing time of 380 s, with learning and detection times of 31 s and 20 s, respectively. LSTM exhibits a processing time of 319 s, 35 s for learning, and 20 s for detection. CNN demands 450 s for processing, 40 s for learning, and 19 s for detection. MM-WMVDEL consumes 500 s for processing, with learning and detection times of 44 s and 21 s, respectively. Impressively, the ROAST-IoT system excels once more, with a processing time of just 200 s, accompanied by 16 s of learning and a rapid detection time of 14 s.

Transitioning to the UNSW-NB 15 dataset, a consistent pattern of efficiency emerges. DNN stands at 550 s for processing, with 19 s for learning and 16 s for detection. Autoencoder requires 500 s for processing, with learning and detection times of 17 s and 15 s, respectively. LSTM exhibits a processing time of 420 s, 23 s for learning, and 20 s for detection. CNN demands 450 s for processing, 25 s for learning, and 23 s for detection. MM-WMVDEL demonstrates processing, learning, and detection times of 500 s, 22 s, and 17 s, respectively. The ROAST-IoT system remains consistently efficient, showcasing a processing time of 310 s, a mere 10 s for learning, and a rapid detection time of just 8 s.

Finally, the Edge IIoT dataset mirrors these trends. DNN requires 650 s for processing, accompanied by 30 s for learning and 17 s for detection. Autoencoder showcases a processing time of 500 s, 27 s for learning, and 18 s for detection. LSTM exhibits a processing time of 460 s, 25 s for learning, and 21 s for detection. CNN demands 550 s for processing, 28 s for learning, and 15 s for detection. MM-WMVDEL stands at 450 s for processing, 18 s for learning, and 16 s for detection. Impressively, the ROAST-IoT system demonstrates its exceptional efficiency once again, boasting a processing time of just 190 s, 14 s for learning, and an astonishingly rapid detection time of 7 s.

In essence, these results collectively underscore the superior efficiency of the ROAST-IoT system in terms of processing, learning, and detection times across various datasets, solidifying its potential as a formidable tool for real-time intrusion detection and response in the realm of IoT security.

## 5. Discussions

The Internet of Things (IoT) has revolutionized numerous industries, offering unparalleled benefits. However, the proliferation and complexity of IoT systems have ushered in a new set of security challenges. In response to these challenges, intrusion detection systems (IDSs) have emerged as critical components for ensuring the security and defense of IoT devices. Leveraging machine learning (ML) techniques for IDSs in IoT networks has gained significant traction in recent times. However, the evolving landscape of IoT environments, characterized by diverse technological and environmental factors, suggests the potential for further development in this field.

It is evident that IoT systems have increasingly adopted ML techniques for IDSs. However, the primary shortcomings of current IoT security frameworks, encompassing limited intrusion detection capabilities, pronounced latency, and protracted processing times, result in undesirable delays. While ML methods have shown promise in IoT security, DL offers distinct advantages in tackling these challenges. DL, a subset of ML, utilizes neural networks with multiple layers to automatically learn complex patterns and representations from data. In the context of IDSs for IoT devices, DL excels due to its ability to handle intricate and high-dimensional data. DL models, such as convolutional neural networks (CNNs) and recurrent neural networks (RNNs), can capture nuanced relationships within IoT data, enhancing intrusion detection accuracy. Furthermore, DL can alleviate latency and processing time concerns through optimized architectures like parallel processing and graphical processing unit (GPU) acceleration. The deep hierarchical features learned by DL models enable faster and more accurate intrusion detection, minimizing the response time to potential threats. In essence, while ML techniques have paved the way for IDSs in IoT security, DL’s capacity to comprehend intricate patterns, coupled with its potential to reduce latency and processing time, positions it as a powerful tool to fortify IoT security frameworks, bridging the gaps posed by inadequate intrusion detection capabilities, latency issues, and processing delays. Table 4 summarizes the above-mentioned comparisons on ML and DL for intrusion detection systems (IDSs) in IoT security.

This work introduces an innovative solution based on DL framework, the range-optimized attention convolutional scattered technique (ROAST-IoT), designed to boost the security of IoT networks against contemporary threats and intrusions. ROAST-IoT employs a strategic approach encompassing the scattered range feature selection (SRFS) model, attention-based convolutional feed-forward network (ACFN) technique, and the modified dingo optimization (MDO) algorithm for loss function estimation. This holistic framework aims to address the intricate challenges posed by IoT security comprehensively. Evaluating the proposed ROAST-IoT system involves rigorous benchmarking against popular intrusion datasets, including ToN-IoT, IoT-23, UNSW-NB 15, and Edge-IIoT. The achieved results underline the effectiveness of the ROAST technique, surpassing the performance of existing cutting-edge intrusion detection systems. Notably, the algorithm demonstrates an impressive accuracy rate of 99.15% on the IoT-23 dataset, 99.78% on the ToN-IoT dataset, 99.88% on the UNSW-NB 15 dataset, and 99.45% on the Edge-IIoT dataset. Moreover, the high area under the receiver operating characteristics curve (AUC-ROC) value of 0.998 showcases the algorithm’s prowess in distinguishing between legitimate data and malicious traffic.

The proposed ROAST-IoT model is further elucidated by delineating its workflow across distinct operational phases. Commencing with intrusion data collection, the model encompasses scattered range feature selection (SRFS) to extract pertinent features and an attention-based convolutional feed-forward network (ACFN) for accurate intrusion classification. The incorporation of the modified dingo optimization (MDO) algorithm contributes to optimal loss function estimation, augmenting the classifier’s precision. Notably, the model’s benefits encompass heightened accuracy, reduced complexity, and minimized time consumption. The presented confusion matrices and receiver operating characteristics (ROC) analyses validate the robustness of the proposed model in accurately classifying intrusions. The inclusion of feature selection mechanisms enhances performance across various datasets. The model’s positive prediction value (PPV), negative prediction value (NPV), sensitivity, and specificity are thoroughly evaluated, demonstrating its superiority over conventional deep learning algorithms. Furthermore, comparative assessments with conventional classification techniques and deep learning models emphasize the superiority of ROAST-IoT. This includes a comprehensive evaluation of accuracy, precision, recall, and f1-score. The model’s exceptional performance is consistently validated across different datasets, underscoring its versatility and efficacy.

In Table 3, each row represents a specific dataset, and for each dataset, different deep learning methods are evaluated for their computational processing times in seconds (s). The methods include deep neural network (DNN), autoencoder, long short-term memory (LSTM), convolutional neural network (CNN), a specific deep learning algorithm (MM-WMVDEL), and the proposed ROAST-IoT system. For example, looking at the first row, for the IoT-23 dataset, the ROAST-IoT system exhibited a computational processing time of 300 s (s), while other methods like DNN, autoencoder, LSTM, and CNN had varying processing times. This comparison provides insights into how efficient the ROAST-IOT system is in terms of computational processing time compared to other deep learning methods across different datasets.

In practice, the proposed ROAST-IoT algorithm showcases a comprehensive and innovative approach for addressing the security challenges inherent in IoT networks. Through a synergistic integration of advanced techniques, including feature selection, deep learning, and optimization, the algorithm excels in intrusion detection accuracy and performance across various datasets. These results underscore its potential to significantly enhance the security posture of IoT systems, providing robust protection against evolving threats and intrusions.

### 5.1. Limitations of Proposed ROAS-IT System

The limitations of the range-optimized attention convolutional scattered technique (ROAST-IoT) are not explicitly mentioned in the provided text. However, based on the general context and challenges associated with developing intrusion detection systems for IoT networks, we can infer potential limitations that may apply to the ROAST-IoT algorithm:Depending on the complexity of the ROAST and the associated machine learning components like scattered range feature selection (SRFS) and attention-based convolutional feed-forward network (ACFN), the algorithm may demand significant computational resources. This could impact its real-time applicability in resource-constrained of IoT devices.Like many machine learning-based intrusion detection systems, the effectiveness of the ROAST-IoT algorithm heavily relies on high-quality, diverse, and representative training data. If the algorithm is sensitive to data quality or distribution, obtaining a suitable dataset could be a challenge.As IoT networks continue to grow in scale and complexity, ensuring the scalability of intrusion detection systems becomes crucial. If the ROAST-IoT algorithm struggles to scale efficiently with the increasing number of IoT devices and network traffic, its practicality may be limited.Intrusion detection systems need to be adaptable to new and evolving attack techniques. If the ROAST-IoT algorithm is not designed to handle emerging threats and variations of existing attacks, its effectiveness could decline over time.Many IoT devices operate with limited computational power, memory, and energy resources. If the ROAST-IoT algorithm requires resource-intensive operations, it may not be suitable for deployment on resource-constrained devices.Achieving a balance between accurately detecting intrusion attacks while minimizing false positives and false negatives can be challenging. If the ROAST-IoT algorithm struggles to achieve this balance, it could lead to either unnecessary alerts or missed detections.IoT networks span various domains, including industrial IoT, healthcare IoT, and smart homes, each with its own unique characteristics and challenges. The ROAST-IoT algorithm’s effectiveness in one domain may not directly translate to another domain.Intrusion detection systems can be susceptible to adversarial attacks aimed at evading detection. If the ROAST-IoT algorithm lacks robustness against such attacks, it may become less effective in a real-world scenario.The limitations of the ROAST-IoT algorithm may be more evident in certain scenarios, network configurations, or types of attacks that have not been fully evaluated in the current study.

### 5.2. Future Works

Based on the information provided about the proposed ROAST-IoT system, the potential avenues for future work and research are as follows:(1)Investigate techniques to enhance the scalability of the ROAST-IoT algorithm. As IoT networks continue to grow in size, the ability to handle a larger number of devices and data flows efficiently is crucial. This could involve exploring distributed computing approaches, optimized data structures, and parallel processing.(2)Develop strategies to optimize the algorithm’s resource usage, making it more suitable for resource-constrained IoT devices. This could involve model compression techniques, efficient feature extraction methods, and algorithmic optimizations to reduce computational requirements.(3)Focus on improving the real-time processing capabilities of the ROAST-IoT algorithm. Achieving low-latency intrusion detection is essential for timely response to threats. Explore techniques to reduce inference time and latency without sacrificing accuracy.(4)Research methods to make the ROAST-IoT algorithm adaptable to emerging threats and attack techniques. This may involve incorporating mechanisms to dynamically update the model based on new attack patterns and vulnerabilities.(5)Investigate techniques to enhance the algorithm’s robustness against adversarial attacks. Adversarial attacks can exploit vulnerabilities in the model, leading to evasion of detection. Implement defenses such as adversarial training or input perturbation to mitigate this risk.(6)Explore the customization of the ROAST-IoT algorithm for specific IoT domains such as industrial IoT, healthcare IoT, or smart cities. Different domains have unique characteristics and challenges, and tailoring the algorithm to these contexts could improve its effectiveness.(7)Research ways to integrate the ROAST-IoT algorithm seamlessly into the overall IoT network architecture. Consider factors such as data preprocessing, communication protocols, and interoperability with other security components.(8)Conduct extensive evaluation of the ROAST-IoT algorithm in real-world IoT environments to assess its performance under diverse conditions. Collaborate with industry partners to deploy and validate the algorithm in practical settings.(9)Investigate approaches that combine the strengths of both machine learning and human experts in intrusion detection. Develop interfaces that enable human analysts to provide feedback, and refine the algorithm’s performance over time.(10)Address privacy concerns that may arise when implementing intrusion detection systems in IoT networks. Explore techniques for secure data sharing and processing while ensuring user privacy.(11)Assess the algorithm’s long-term performance and adaptability as the threat landscape and IoT ecosystem evolve over time.

By pursuing these directions, researchers can further enhance the capabilities, applicability, and robustness of the ROAST-IoT algorithm, contributing to the field of IoT security and intrusion detection.

## 6. Conclusions

An effective artificial intelligence model called ROAST-IoT is presented in this paper for successful identification of intrusions in the IoT context. The proposed framework makes use of a multi-modal design to successfully capture the intricate connections between various kinds of network traffic data. Sensors first capture the network system’s behavior, which is then stored on a cloud server for analysis. Here, available benchmark datasets like IoT 23, Edge-IIoT, ToN-IoT, and UNSW-NB 15 are used for system assessment and evaluation. The state-of-the-art SRES technique is then utilized to choose the most important and necessary features from the available data after data collection. This phase includes the processes of class vector construction, scatter matrix estimation, feature ranking, and feature vector formation. At the end of the procedure, the set of chosen features is then provided to the ACFN classifier for intrusion recognition and classification. The complex deep learning architecture incorporates the features of convolution and feed-forward network models. With the use of the MDO technique, the loss function estimation is carried out as efficiently as possible in this mechanism, enabling the classifier to forecast accurate results. The accuracy of the ROAST-IoT model in predicting different types of attacks was demonstrated by the confusion matrices for the UNSW-NB 15 and IoT-23 datasets. ROC analysis was also conducted on the IoT-23, ToN-IoT, and UNSW-NB 15 datasets, and the ROC curves and AUC values showed the model’s ability to distinguish between different classes of attacks. When comparing the ROAST-IoT model with other deep learning techniques such as CNN-LSTM, CNN-BiLSTM, CNN-GRU, and others, it is found that the proposed model is superior. It is worth noting that the ROAST-IoT system consistently demonstrated shorter processing, learning, and detection times when compared to other methods.

Overall, the higher system correctness, reduced complexity, and reduced time consumption are the main advantages of employing the proposed ROAST-IoT architecture. For system validation and study, the different types of parameters including PPV, NPV, precision, accuracy, detection rate, f1-score, etc., have been computed in this study using a variety of datasets. The final comparison outcomes indicate that the proposed ROAST-IOT model outperforms other classification techniques with superior results. In future, the current study can be further enhanced with the use of some other deep learning (DL) model for protecting IIoT networks against cyberattacks.

## Figures and Tables

**Figure 1 sensors-23-08044-f001:**
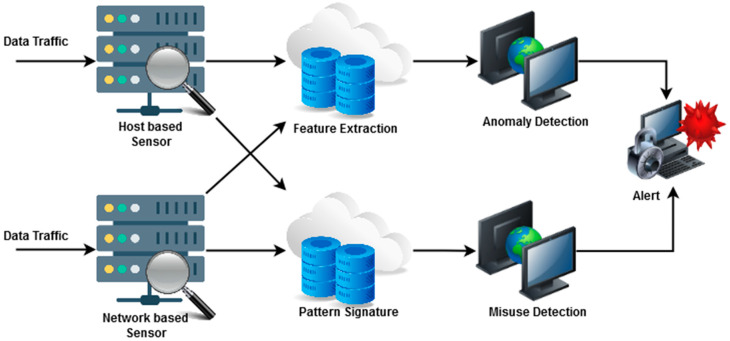
A visual structure of a typical intrusion detection system (IDS) model.

**Figure 2 sensors-23-08044-f002:**
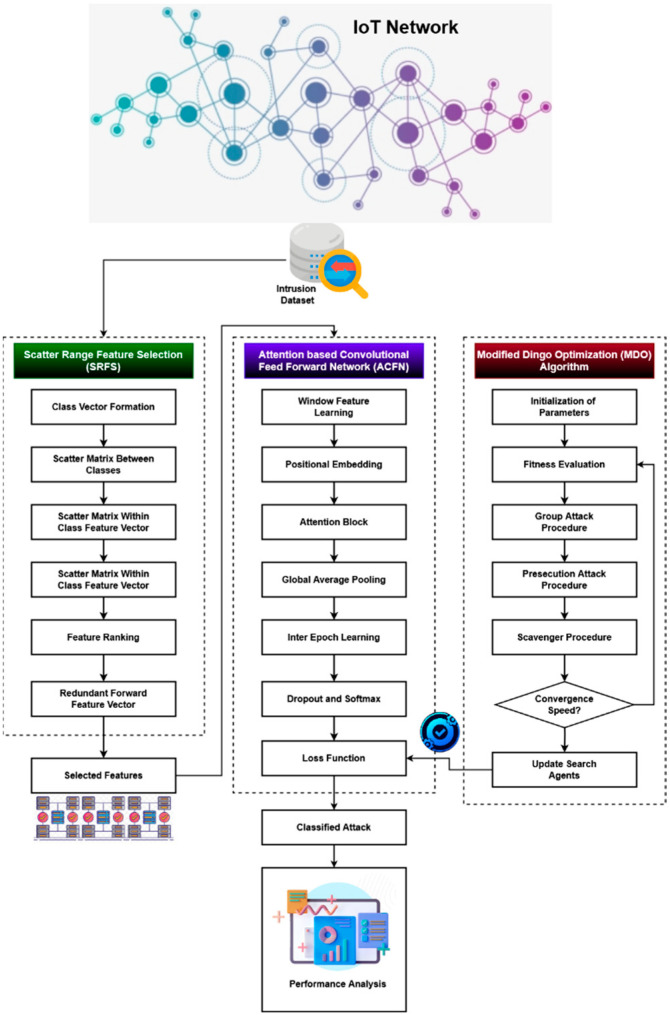
A systematic flow diagram of the proposed ROAST-IoT framework.

**Figure 3 sensors-23-08044-f003:**
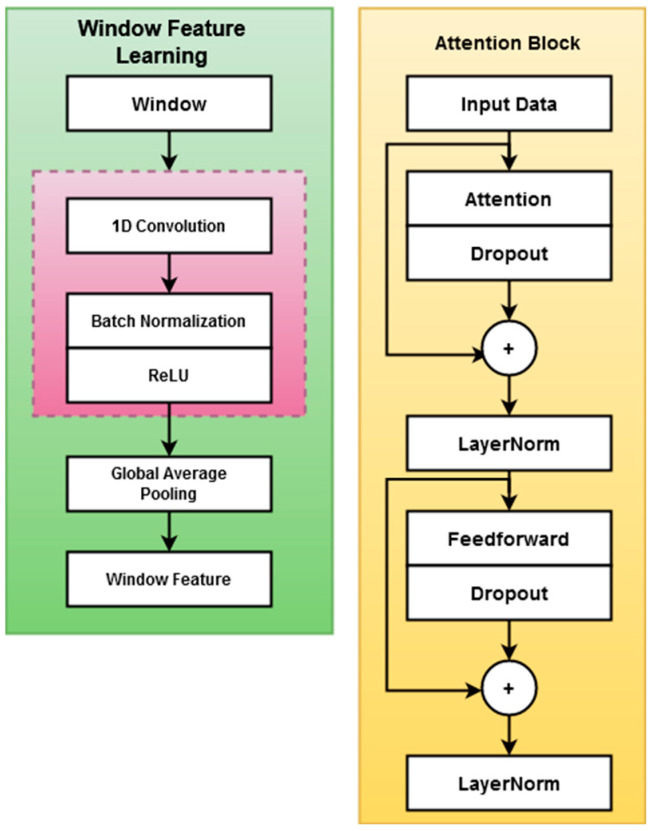
Attention-based convolutional feed-forward network (ACFN) classification model.

**Figure 4 sensors-23-08044-f004:**
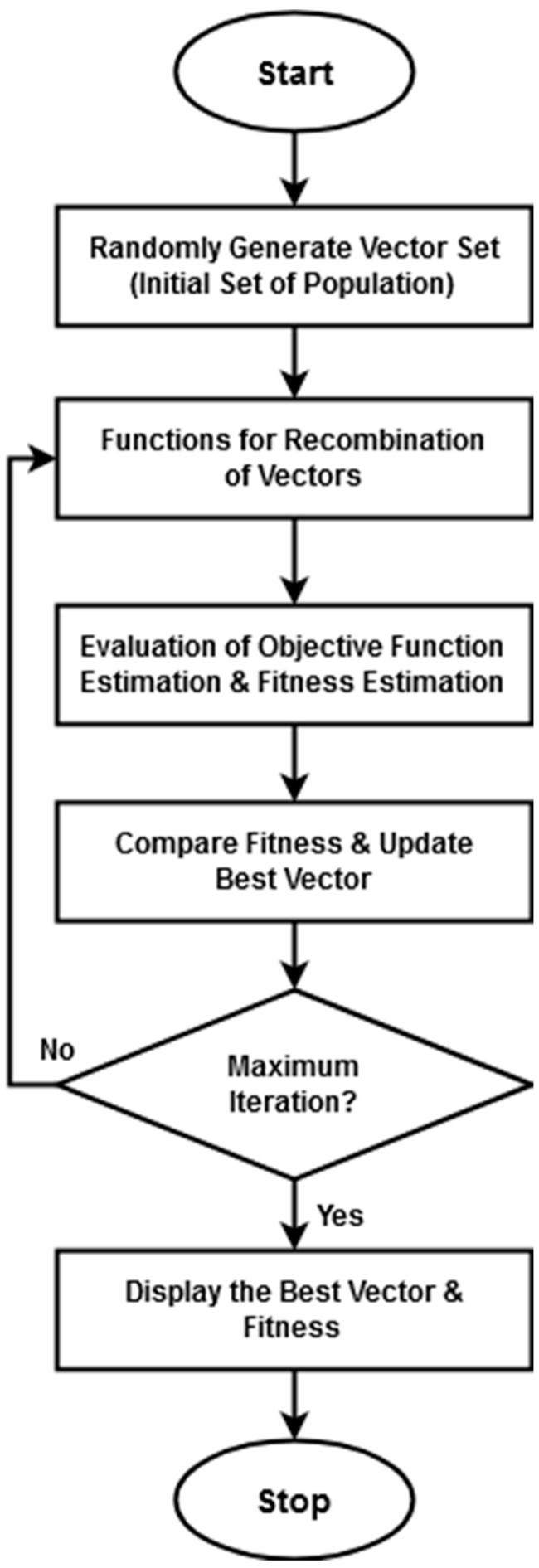
Flow diagram of proposed modified dingo optimization (MDO) algorithm.

**Figure 5 sensors-23-08044-f005:**
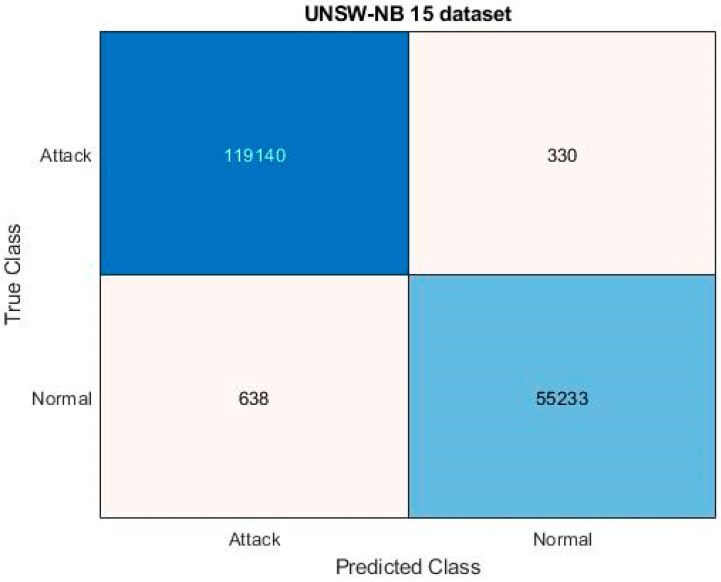
Confusion matrix for proposed system with attack and normal categories on UNSW-NB 15 dataset.

**Figure 6 sensors-23-08044-f006:**
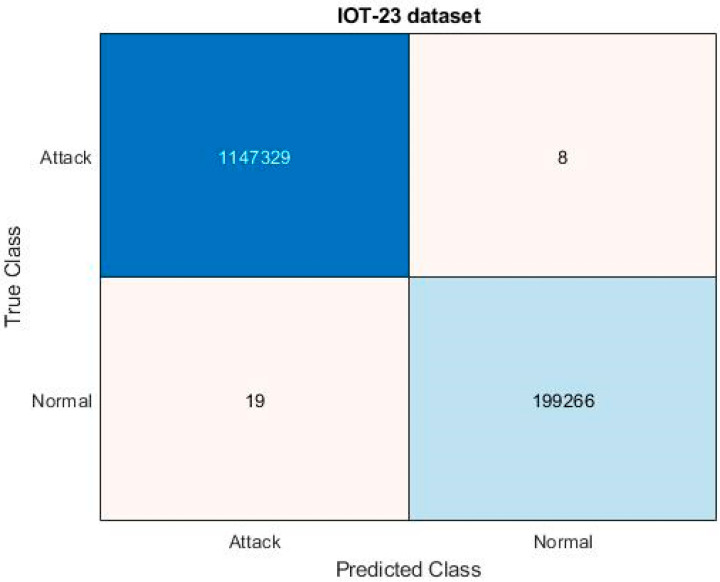
Confusion matrix for proposed system with attack and normal categories on IOT-23 dataset.

**Figure 7 sensors-23-08044-f007:**
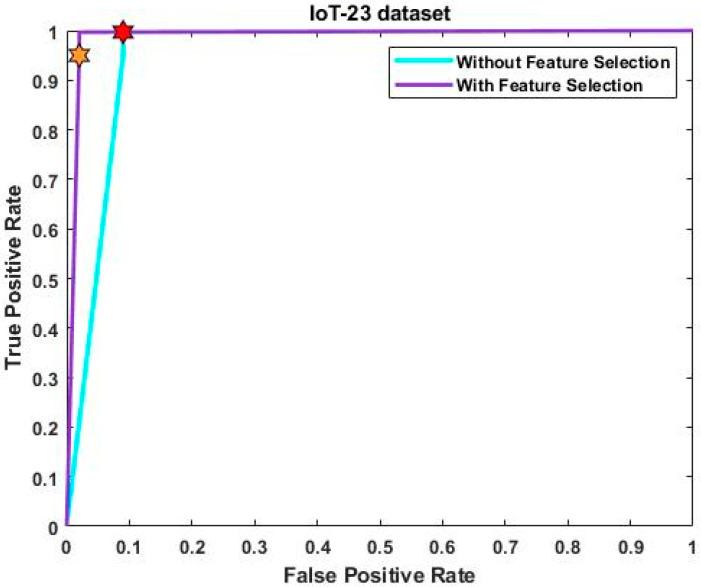
ROC analysis with and without feature selection of IoT-23 dataset.

**Figure 8 sensors-23-08044-f008:**
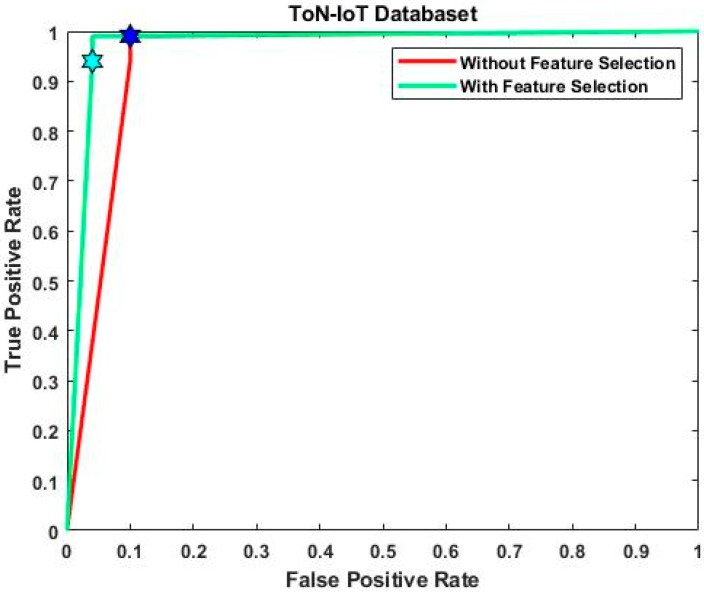
ROC analysis of ToN-IoT dataset with and without feature selection.

**Figure 9 sensors-23-08044-f009:**
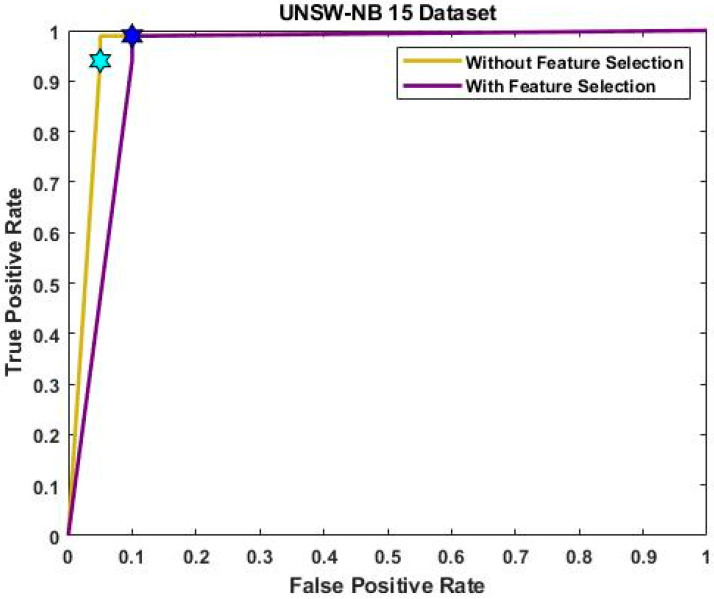
ROC analysis of UNSW-NB 15 dataset with and without feature selection.

**Figure 10 sensors-23-08044-f010:**
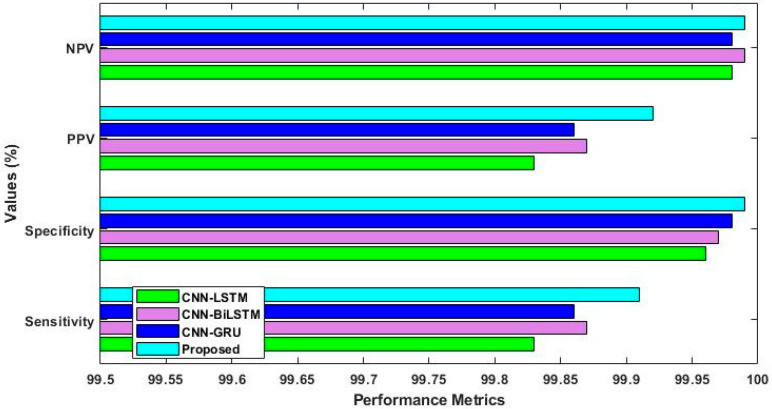
Performance comparison with other deep learning techniques.

**Figure 11 sensors-23-08044-f011:**
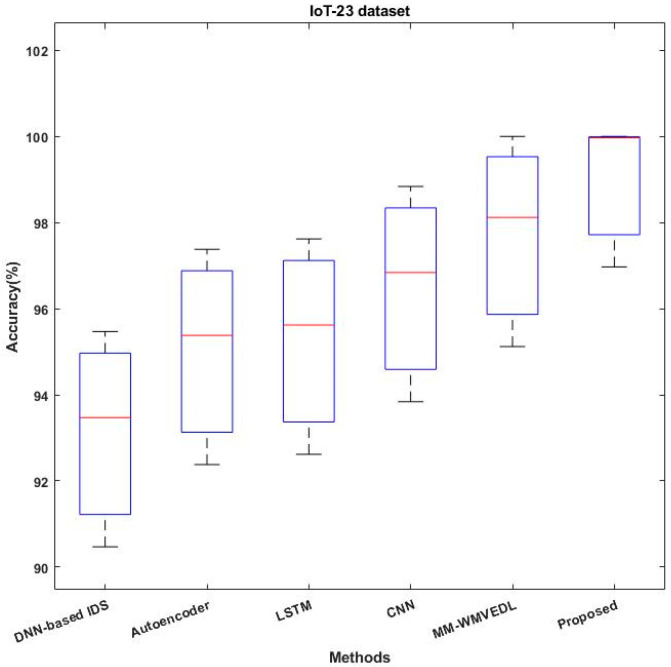
Proposed system accuracy analysis using IoT-23 dataset.

**Figure 12 sensors-23-08044-f012:**
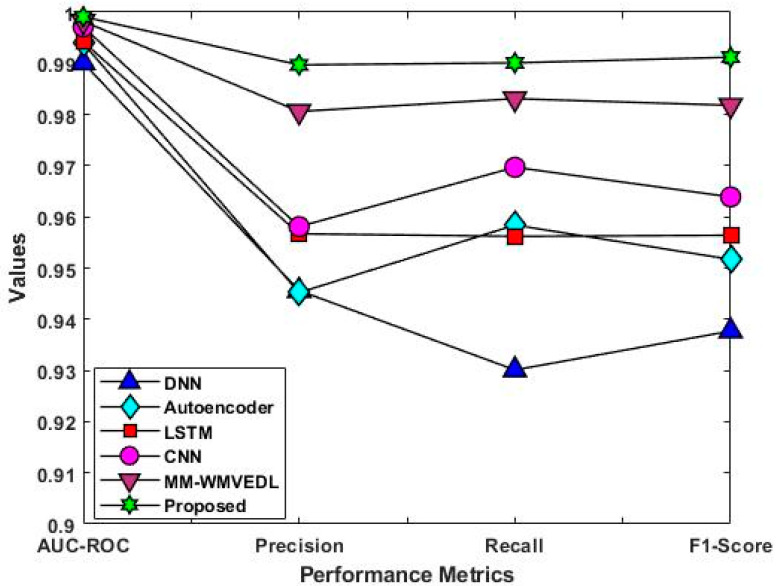
Overall performance analysis using IoT-23 dataset using various deep learning algorithms compared to proposed system.

**Figure 13 sensors-23-08044-f013:**
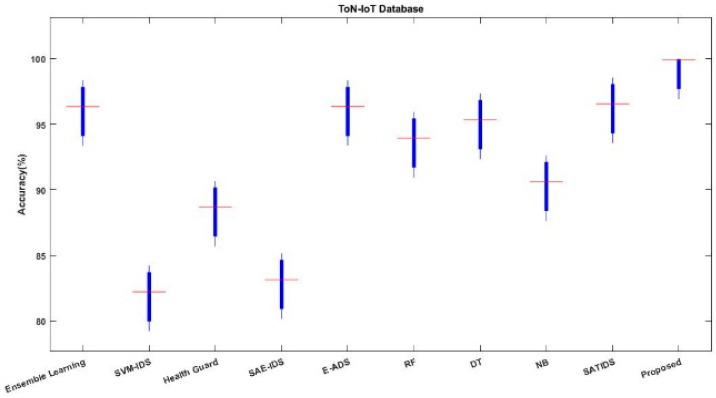
Accuracy analysis using ToN-IoT dataset.

**Figure 14 sensors-23-08044-f014:**
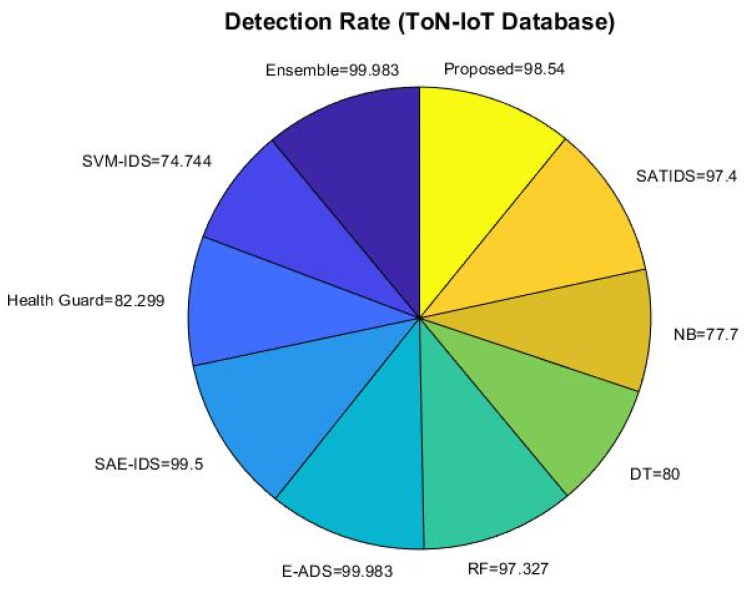
Detection rate using ToN-IoT dataset.

**Figure 15 sensors-23-08044-f015:**
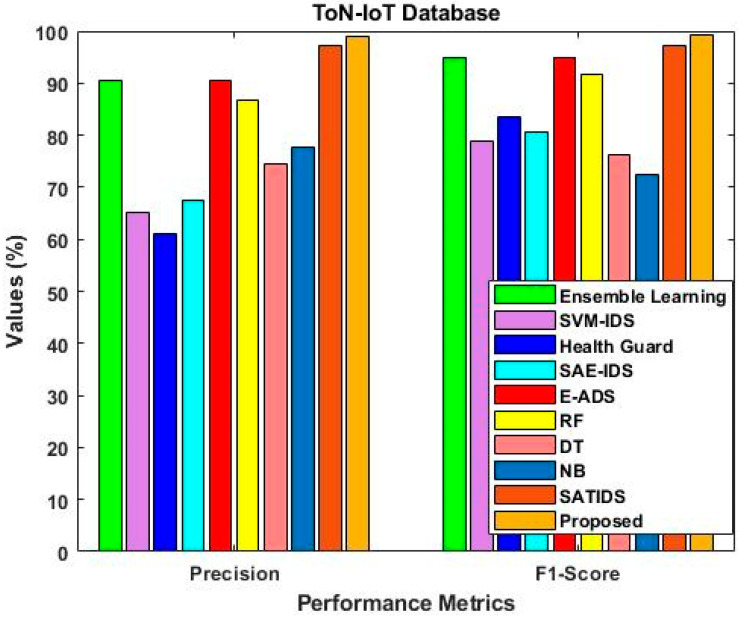
Comparative analysis using ToN-IoT dataset.

**Figure 16 sensors-23-08044-f016:**
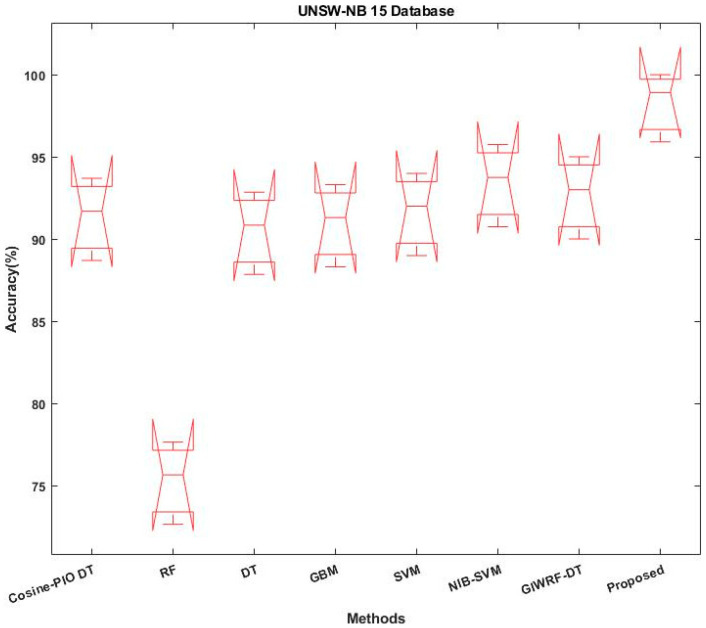
Accuracy analysis using UNSW-NB 15 dataset.

**Figure 17 sensors-23-08044-f017:**
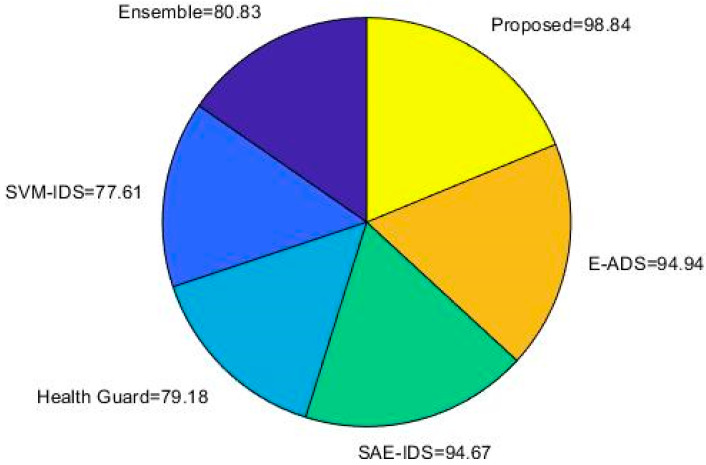
Accuracy using Edge-IIoT dataset.

**Figure 18 sensors-23-08044-f018:**
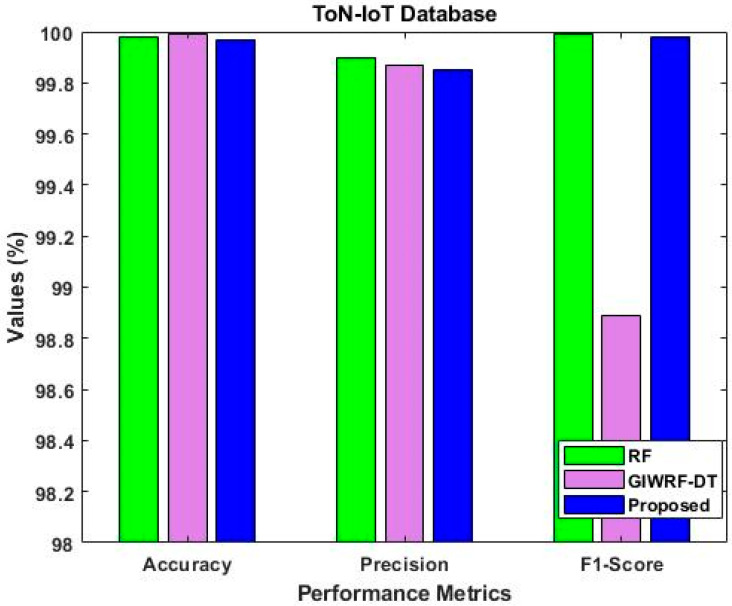
Performance comparison using ToN-IoT dataset.

**Figure 19 sensors-23-08044-f019:**
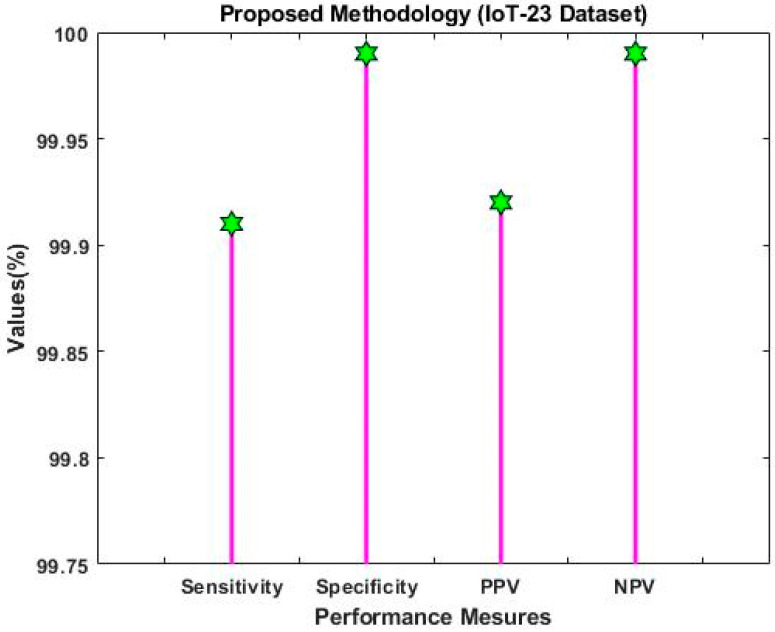
Overall performance analysis using IoT-23 dataset.

**Table 1 sensors-23-08044-t001:** List of online available datasets used for IoT security.

No.	Datasets	Different Attack Scenarios	Heterogeneity of IoT Data Sources	Number of Instances	Number of Features
1	KDD’99	No	No	494,021	41
2	NSL-KDD	No	No	125,973	42
3	UNSW-NB 15	Yes	No	2,540,000	49
4	LWSNDR	No	No	48,000	29
5	ISCX	Yes	No	15,570	45
6	AWID	Yes	No	458,691	155
7	UNSW-IoT trace	No	No	1,000,000	8
8	BoT-IoT	Yes	No	56,800	116
9	T-IIoT	Yes	Yes	50,000	52

**Table 2 sensors-23-08044-t002:** A table summarizing the state-of-the-art studies from the literature review, including the approach, key contributions, and limitations of each study.

Reference	Approach	Key Contributions	Limitations
Da Costa et al. [18]	Review of machine learning techniques for IoT intrusion detection	Comprehensive review of IoT intrusion detection techniques	High false positive rate across research studies
Islam et al. [19]	Use of machine learning and deep learning algorithms for IoT threat detection	Focus on quick implementation and effective handling of unknown events	Limited discussion on computational efficiency
Nimbalkar et al. [20]	Feature selection techniques for enhancing IDS in IoT networks	Effective reduction of feature count for accurate detection	Limited discussion on scalability for large datasets
Hindy et al. [21]	Case study on machine learning techniques for intrusion detection	Focus on MQTT-IoT-IDS 2020 dataset and bi-directional features	No explicit mention of limitations in the provided text
Alsaedi et al. [22]	Data-driven approaches for IoT and IIoT intrusion detection	Development of new dataset for IoT intrusion detection	Limited information on performance evaluation
Zhou et al. [23]	GNN-based intrusion detection using HAA and RWR techniques	Innovative use of GNN for unknown attack detection	Increased time consumption for predictions
Wahab et al. [24]	Online deep learning approach with data drift detection	Detection of intrusion data streams with drift detection	High computational and time complexity
Kan et al. [25]	Adaptive PSO-CNN model for multi-type intrusion detection	Incorporation of PSO with CNN for improved reliability	Limited details on how cross-entropy loss is implemented
Abdalgawad et al. [26]	Bi-GAN model for detecting cyberattacks in IoT networks	Effective use of Bi-GAN and generative deep learning	No specific limitations provided in the text
Kumar et al. [27]	Distributed intrusion detection for DDoS attacks in IoT	Development of distributed intrusion detection system	Limited details on the specific methodology used
Shukla et al. [28]	AI-based intrusion detection in IoT with focus on data selection	Examination of various machine learning methods in IoT	No explicit discussion of limitations in the provided text

**Table 3 sensors-23-08044-t003:** Computational processing time for other deep learning algorithms compared to ROAST-IoT system.

No.	Datasets	Methods	Processing time	Learning Time	Detection Time
1.	IoT-23	DNN	450 (s)	25 (s)	20 (s)
		Autoencoder	400 (s)	22 (s)	18 (s)
		LSTM	379 (s)	18 (s)	16 (s)
		CNN	400 (s)	24 (s)	21 (s)
		MM-WMVDEL	550 (s)	26 (s)	24 (s)
		ROAST-IOT	300 (s)	12 (s)	4 (s)
2.	ToN-IoT	DNN	400 (s)	35 (s)	30 (s)
		Autoencoder	380 (s)	31 (s)	20 (s)
		LSTM	319 (s)	35 (s)	20 (s)
		CNN	450 (s)	40 (s)	19 (s)
		MM-WMVDEL	500 (s)	44 (s)	21 (s)
		ROAST-IOT	200 (s)	16 (s)	14 (s)
3.	UNSW-NB 15	DNN	550 (s)	19 (s)	16 (s)
		Autoencoder	500 (s)	17 (s)	15 (s)
		LSTM	420 (s)	23 (s)	20 (s)
		CNN	450 (s)	25 (s)	23 (s)
		MM-WMVDEL	500 (s)	22 (s)	17 (s)
		ROAST-IOT	310 (s)	10 (s)	8 (s)
4.	Edge IIoT	DNN	650 (s)	30 (s)	17 (s)
		Autoencoder	500 (s)	27 (s)	18 (s)
		LSTM	460 (s)	25 (s)	21 (s)
		CNN	550 (s)	28 (s)	15 (s)
		MM-WMVDEL	450 (s)	18 (s)	16 (s)
		ROAST-IOT	190 (s)	14 (s)	7 (s)

**Table 4 sensors-23-08044-t004:** A simplified table comparing machine learning (ML) and the benefits of deep learning (DL) for intrusion detection systems (IDSs) in IoT security.

Aspect	Machine Learning (ML)	Deep Learning (DL)
Data Complexity	Effective for simple data representations.	Proficient in handling complex, high-dimensional data.
Intricacy Handling	Limited ability to capture intricate patterns.	Excels in capturing intricate relationships and patterns.
Model Architecture	Relies on conventional algorithms and methods.	Utilizes deep neural networks with multiple layers.
Intrusion Detection Accuracy	May struggle with nuanced threat detection.	Enhanced accuracy due to hierarchical feature learning.
Latency and Processing Time	Latency and processing time can be noticeable.	Optimized architectures reduce latency and processing time.
Response to Threats	Response time might be longer due to processing.	Faster response to threats with quicker analysis.
Aspect	Machine Learning (ML)	Deep Learning (DL)
Data Complexity	Effective for simple data representations.	Proficient in handling complex, high-dimensional data.

## Data Availability

Not applicable.

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
