# Peer review of "ROAST-IoT: A Novel Range-Optimized Attention Convolutional Scattered Technique for Intrusion Detection in IoT Networks"

_sensors, 2023, doi:10.3390/s23198044_

Round 1
Reviewer 1 Report
The paper is well written, tackles an important and relevant topic to IoT, and had solid technical foundations. Below are just minor observations to make it even better.
Technical notes
At the system operation level, each node will communicate its data to the cloud, that will run the pre-trained model and detect if there is an intrusion attach or not. Right, if this is the case, then will there be any concern regarding the IoT nodes’ energy consumption especially that in many cases, these nodes may not have a permanent energy sources and may operate on batteries. Please provide more discussion highlighting this challenge and how it can be solved/mitigated?
Language and format
- The acronyms should be defined ONLY once at the first time they appear in the paper, then either the acronyms or the full name should be used later. Make sure that this is the case. For example IoT, SRFS, MDO ACFN are defined more than once in the paper.
- Some words are capitalized for no reason (e.g. Scenarios and Sources in Table 1)
- Figures 1 and 2 captions should be below the figure itself
- I suggest using the math style for all equations and variables,
- After Eq.1 I think there is a small mistake in the variable X_ic’^F, there should be no ‘ above the letter c.
Author Response
Original Manuscript ID: ID: sensors-2619651
Original Article Title: ROAST-IoT: A Novel Range Optimized Attention Convolutional Scattered Technique for Intrusion Detection in IoT Networks
To: Editor in Chief,
MDPI, Sensors
Re: Response to reviewers
Dear Editor,
Many thanks for insightful comments and suggestions of the referees. Thank you for allowing a resubmission of our manuscript, with an opportunity to address the reviewers’ comments.
We are uploading (a) our point-by-point response to the comments (below) (response to reviewers), (b) an updated manuscript with green, blue, and orange highlighting indicating changes, and (c) a clean updated manuscript without highlights (PDF main document).
By following reviewers’ comments, we made substantial modifications in our paper to improve its clarity, English and readability. In our revised paper, we represent the improved manuscript such as:
(1) Revised Abstract, (2) Revised Introduction, (3) Results section, (4) Discussions and Conclusion sections.
We have made the following modifications as desired by the reviewers:
Best regards,
Corresponding Author,
Dr. Qaisar Abbas (On behalf of authors),
Professor.

Reviewer 2 Report
Paper on "ROAST-IoT: A Novel Range Optimized Attention Convolutional Scattered Technique for Intrusion Detection in IoT Networks" is good read and it can be accepted after following queries:
1. Why Dingo Optimization (MDO) algorithm is used and how weights can enhnace the accuracy?
2. What is the purpose of the different layers of CNN.\?
3. Few more papers can be considered for reference like 1. Secure hierarchical fog computing based architecture for industry 5.0 using an attribute-based encryption scheme, and Detection of Android Malware in the Internet of Things through the K-Nearest Neighbor Algorithm
4. Analysis of data sets for True positive is shown in different figures. Add one more figure and you can use pi-chart for the same where all datasets can be shown.
Author Response

(The authors gave the same response as above.)

Reviewer 3 Report
The manuscript gives a condensed overview of an Intrusion Detection Systems (IDS) method for Internet of Thinks (IoT) systems using the newly implemented Range Optimised Attention Convolutional Scattered Technique (ROAST-IoT).
The manuscript is well structured and contains a clear introductory chapter on the need for a well-functioning IDS for the IoT domain. Chapter 2 provides a literature and test data set overview of ongoing and completed research activities. Chapter 3 includes a detailed description of the proposed ROAST-IoT implementation and presents the work results in Chapter 5. The manuscript is concluded with a summary in the final Chapter 6.
The manuscript requires in-depth knowledge of different data analysis principles for machine learning (ML) and deep learning (DL). The reader must have a deep understanding of the different types of neural networks and their mathematical descriptions.
I have the following comments:
· The parameters used in the mathematical descriptions are only partially described. A list of the parameters used at the end of the document would be very helpful.
· Check the numbering of the mathematical formulas. Especially see the numbering after equation 16 and after equation 20.
· Include the long versions of the abbreviations used, e.g.: for SVM, DT, RF, LSTM and DBN (see page 4 below).
· Reference all figures in the text (for example: Figure 2, 4, 7, 8, 9, 14, 15 and17). If necessary, add a one sentence description.
The manuscript is written in a well readable and understandable manner. For a complete understanding in detail, the reader must have specialist knowledge.
References are checked only randomly. All tested references were found.
The manuscript can be, from my point of view, released, after a minor revision, for publication.
Author Response

(The authors gave the same response as above.)

Reviewer 4 Report
The paper proposes a novel Range Optimized Attention Convolutional Scattered Technique (ROAST-IoT) to protect IoT networks from modern threats and intrusions.
A minor revision is required.
Strengths: ROAST-IoT algorithm efficiency, method to classify attacks, comparison of performances of different machine learning methods.
Points of weakness: few discussions.
Actions to do:
According to the weaknesses, I suggest to improve the paper by answering to these points:
· Please add more details of the dataset (Table 1).
· Conclusion section should be improved;
· Further references should be added in the introduction section about AI applied for intrusion attacks in different applications such as:
- https://doi.org/10.3390/s22041494
- doi: 10.1109/ACCESS.2020.3034399
- https://doi.org/10.3390/electronics11142180
- https://doi.org/10.3390/app13137507
- https://doi.org/10.3390/sym12050754
Minor remarks:
Please add more comments in figure captions.
Minor editing of English language required.
Author Response

(The authors gave the same response as above.)
